# High risk of unprecedented UK rainfall in the current climate

Vikki Thompson[1], Nick J. Dunstone[1], Adam A. Scaife[1], Doug M. Smith[1], Julia M. Slingo[1], Simon Brown[1] & Stephen E. Belcher[1]

In winter 2013/14 a succession of storms hit the UK leading to record rainfall and flooding in many regions including south east England. In the Thames river valley there was widespread flooding, with clean-up costs of over £1 billion. There was no observational precedent for this level of rainfall. Here we present analysis of a large ensemble of high-resolution initialised climate simulations to show that this event could have been anticipated, and that in the current climate there remains a high chance of exceeding the observed record monthly rainfall totals in many regions of the UK. In south east England there is a 7% chance of exceeding the current rainfall record in at least one month in any given winter. Expanding our analysis to some other regions of England and Wales the risk increases to a 34% chance of breaking a regional record somewhere each winter.

[1] Met Office Hadley Centre, Exeter EX1 3PB, UK. Correspondence and requests for materials should be addressed to V.T. (email: vikki.thompson@metoffice.gov.uk)

In the winter of 2013/2014 an exceptional series of storms affected the UK[1-4]. In January 2014 south east England experienced unprecedented rainfall 30% higher than any previous January for over a century[5-7] and the River Thames remained exceptionally high for longer than any previous flooding event since records began in 1883[8-10]. The rainfall of winter 2013/2014 had large impacts on infrastructure and businesses. The record rainfall led to 18,700 insurance claims related to flooding across the UK, costing an estimated £451 m[11].

Unprecedented rainfall events have also recently occurred in other regions. In northern England 2015 was the wettest December on record[7, 12]. October 2000 was the wettest on record across England and caused widespread flooding[13, 14]. Understanding the probability of extreme rainfall and how extremes could affect local hydrological conditions is essential for policy makers, contingency planners and insurers.

Prior to these events there was no direct observational evidence for such high rainfall totals—quantifying the chance of extreme rainfall is fundamentally constrained by the limited length of the recent observational record. However, climate models can provide a much larger sample of events that are meteorologically plausible, potentially providing a more realistic estimate of the risk of extremes[15, 16]. Here we present the UNSEEN method—UNprecedented Simulated Extremes using ENsembles. We use the Met Office near term climate prediction system to provide multiple simulations of the current climate[17]. The system uses the Hadley Centre global climate model, HadGEM3-GC2[18], at high resolution compared to most current climate prediction models: 60 km atmosphere and 0.25° ocean. The model is initialised with atmospheric, oceanic, and sea-ice observational data and current anthropogenic and natural forcings, so that the simulations are representative of current real world climate. Using the large ensemble of simulations from decadal climate prediction studies provides considerably more realisations than are available from the recent observational period. In this case the model provides over 100 times more winters than have been observed over the current climate period 1981–2015 ('Methods'). The model is therefore capable of directly sampling more extreme cases than the available observations, allowing the identification of unprecedented rainfall events to assess their likelihood in the real world.

We find many unprecedented monthly rainfall totals for south east England in the model simulations. Examples of the atmospheric circulation patterns for several extreme rainfall months are assessed, showing a variety of conditions can lead to high rainfall totals. For south east England we find a 7% chance of a rainfall total greater than the current observed record in at least one month of a given winter.

## Results

**January rainfall record.** The observed and modelled January rainfall totals for south east England are shown in Fig. 1. Prior to January 2014, observations do not provide a precedent for the totals seen in 2014. The observed rainfall and two examples of model simulations are shown in Fig. 1a. Both model simulations show variability that appears similar to the observations. One set of model simulations shows no rainfall totals outside of the spread of observations. However, the other shows a month with rainfall far greater than had yet been observed—and even greater than was observed in the current record month of January 2014.

The distribution from all 4200 modelled January monthly rainfall totals is shown in Fig. 1b alongside the observed January rainfall. There are many model months with greater rainfall than had been observed prior to 2014, and several even exceed the January 2014 total and are therefore unprecedented. This shows that the level of rainfall seen in January 2014 could have been expected and these simulations could be used to calculate chance of exceeding current record monthly rainfall totals, assuming the climate is stationary over the observational period used.

**Model fidelity.** In order to accurately calculate the chance of unprecedented monthly rainfall, the model must be able to provide a realistic representation of the range of states available to the real world[19]. While this high-resolution climate model is able to represent weather patterns that lower resolution models struggle to simulate, e.g. Atlantic blocking[20], it is still not perfect and is unable to fully represent fine scale orographic enhancement that often gives rise to flooding for example[21]. However, over many regions, including south east England, model error due to orography is small. We apply tests to ensure that the model represents observations accurately enough to look for unprecedented rainfall events.

To assess the model fidelity we use the Met Office NCIC data set[7] covering 1981–2015, the same period as the model simulations. Later we discuss the use of longer observational data sets. The modelled and observed distribution of the monthly rainfall totals for the winter months, October–March, for south east England are statistically indistinguishable (Fig. 2a). We assess the mean, standard deviation, skewness and kurtosis ('Methods').

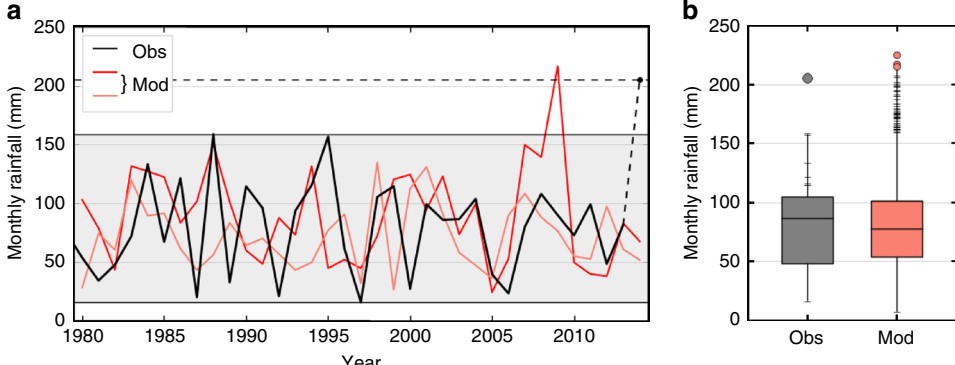

**Fig. 1** Unprecedented monthly rainfall. **a** South east England January monthly rainfall totals from 1979 to 2014, in mm (*black*) and two examples from model simulations (*red*). The *black dashed line* indicates the unprecedented January 2014 observed rainfall total. **b** Distribution of south east England January monthly rainfall totals from observations (*grey*) and the model (*red*). The *box* represents the interquartile range and the range of the *whiskers* represents the minimum and maximum monthly rainfall totals. The *grey dot* indicates the record observed monthly rainfall of January 2014 and *ticks* on the upper end of observations show the values in the upper quartile of previous events. *Ticks* on the model line indicate months above the observed record prior to 2014, and *red dots* indicate even higher totals

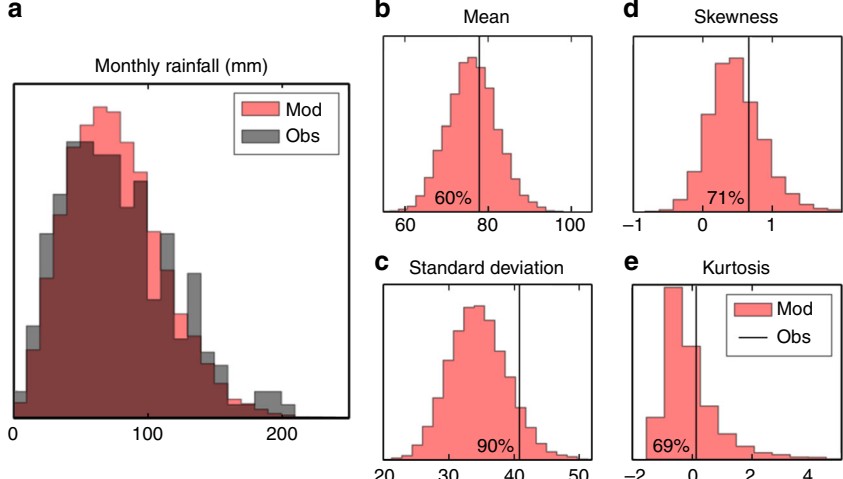

**Fig. 2** Model and observed rainfall totals are indistinguishable. **a** The distribution of October to March monthly rainfall totals (in mm) of observations from the Met Office NCIC data set, 1981 to 2015, and the model simulations for south east England. **b–e** The distributions of proxy south east England model time series mean, standard deviation, skewness and kurtosis compared to the observed values indicated by the *black vertical line*

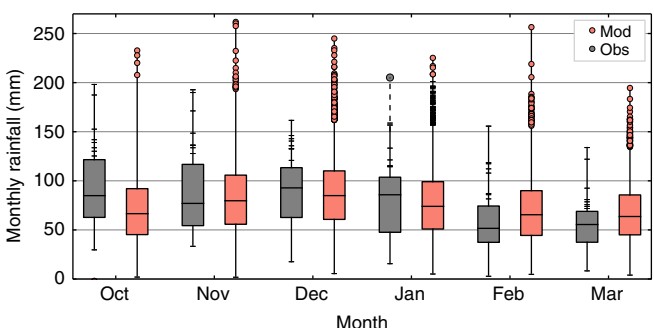

**Fig. 3** Unprecedented monthly rainfall in all winter months. South east England monthly rainfall totals from observations (*grey*) and the model (*red*) for October to March. The *box* represents the interquartile range and the range of the *whiskers* represents the minimum and maximum monthly rainfall totals. *Red dots* indicate model months with greater total rainfall than has yet been observed and *ticks* on the upper observations line indicate values in the upper quartile of events. For January the *ticks* on the model line indicate months above the observed record prior to 2014 and the *grey dot* above the observations indicates the record observed monthly rainfall of January 2014

The skewness and kurtosis measure the shape of the distribution —the symmetry and weighting of the tails. If any of the four measures were unrealistically high in the model then too many unprecedented extremes would be identified, leading to an overestimation of the likelihood; if they are too low then the likelihood would be underestimated. As we are looking for events in the extremes of the distribution it is important to ensure the behaviour in the model tails is indistinguishable from the observations. Proxy time series are generated by sampling the climate model data for sets of equal length to the observed record. The observed value is then compared to the distribution of values from the proxy time series. In south east England the observed values all lie within 95% of the model distribution (Fig. 2b–e) and hence the model is deemed to be statistically indistinguishable from the observations.

**Risk of a monthly rainfall record**. In January 2014 the observed monthly rainfall record was exceeded by 30%[7]. The model

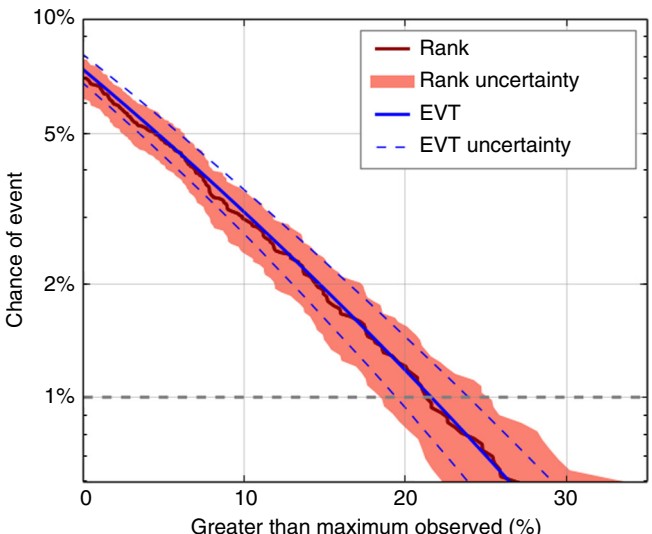

**Fig. 4** Improved risk estimates of the chance of unprecedented rainfall. The chance of an event exceeding the observed maximum monthly rainfall total in south east England in winter (October to March) of any given year from two methods, ranking (*red*) and extreme value theory (*blue*) on the model simulations. The uncertainties indicate the 95% range

simulations (Figs. 1b, 3) show that there was a 7% chance of an unprecedented rainfall total in a given winter, suggesting that the January 2014 rainfall could have been anticipated under the current climate, with the exact timing and occurrence determined simply by natural variability. The number of unprecedented monthly rainfall totals shown by the model varies between months, with December showing the largest number of unprecedented rainfall events (Fig. 3). This suggests that the observed record for December has been fortuitously low, and that the south east of the UK has been fortunate in not experiencing a wetter December in the past 35 years. Considering the whole winter, the chance of a record rainfall in any month in south east England is 7%; therefore, in the current climate, a new record is likely to occur there in the next decade. January 2014 is the wettest month in the observational record but even this event is exceeded by the model, suggesting that even greater rainfall is possible.

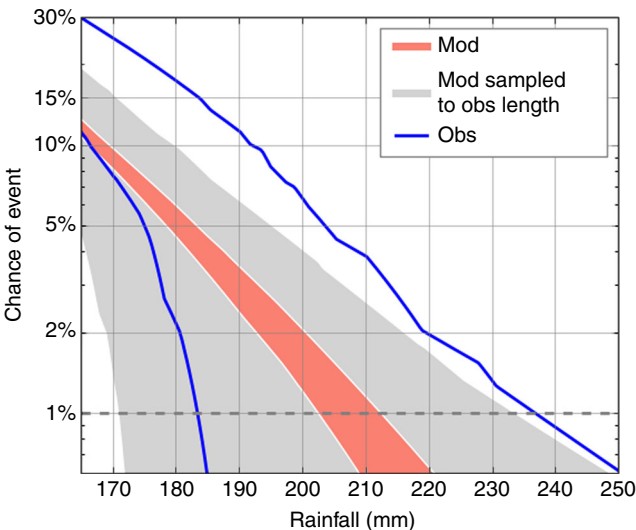

**Fig. 5** The chance of specified absolute rainfall totals. The chance, in south east England, of at least one month in a given winter (October–March) exceeding absolute rainfall totals (in mm) using extreme value theory. The 95% uncertainty cone of the model using the full model data is shown in *red*. Limiting the sample length to 35 years, as is available from observations, is shown in *grey*. The limits of the uncertainty range from the observations alone are shown in *blue*

A statistical model could be used to calculate the probability of unprecedented rainfall from observational data alone using extreme value theory[22–24]. However there are many advantages of using a dynamical model. The observations are poorly sampled at the tails due to the limited length of the record; simply extrapolating from the available observations using extreme value theory would give large errors, as explored later. A statistical model can also lead to physically unrealistic results whereas a dynamical model provides physically plausible limits, but a statistical model may reduce sampling uncertainties by providing a parametric description. In this study both a statistical approach and a dynamical model are used. As the dynamical model provides global fields of many climatic indicators it allows calculation of the likelihood of concurrent climatic events in different locations, for example, the associated storm depths or extreme wind speeds. A dynamical model also allows investigation of the mechanisms of extreme events and remote precursors, which can aid prediction. This is discussed further in the next subsection.

We now quantify the probability of unprecedented rainfall in south east England as an example of our analysis (Fig. 4). Using the dynamical model rainfall data we use both a ranking method and extreme value theory to estimate the chance of record rainfall ('Methods'). Both methods are in good agreement, indicating a 7% risk of a record monthly rainfall total in any winter month of a given year, as we previously noted. The horizontal line at 1% indicates an event with a 1% annual probability would be 20–25% wetter that the current observed monthly record. Note, for south east England longer observational records are available; this is not true for all regions globally. We have also assessed the risk using the longer time series from 1880 to 2015, containing 100 times fewer station records[25], finding a likelihood for an unprecedented rainfall total in at least one month of a given winter of 6%—similar to the values using observations from the model period alone.

Engineers and hydrologists are likely to need the absolute magnitude of extreme events for design purposes. Figure 5 shows the chance of absolute monthly rainfall totals, calculated from both the model simulations and the observations. For the model we have calculated the uncertainty using the full model data

(Fig. 5, *red*). By subsampling the model to the length of observations, 35 years (Fig. 5, *grey*), we show how uncertain results calculated using the observations alone are. This can be compared to the parametric uncertainty range calculated from observations (Fig. 5, *blue*), showing that the spread is similar to the subsampled model curve. The model indicates a slightly lower chance of exceeding the record than observations. From the upper limits of the subsampled model curve (Fig. 5, *grey*) there is a 1% chance of at least 234 mm precipitation in at least one month of a given winter, using all available model data the chance of rainfall at this level is much lower with only two events above this threshold—a 0.05% chance. The figure shows that using the large ensemble of model simulations we are able to greatly constrain the uncertainties compared to using observations alone. We can use Fig. 5 to assess the chance of different precipitation totals. The model simulations suggest a 1% annual probability of monthly rainfall of 205–210 mm. In January 2014 south east England experienced record monthly rainfall of 205 mm; Fig. 5 suggests the chance of such event is just over 1% per year.

**Dynamics of extreme rainfall**. The dynamical model provides global fields of dynamically consistent variables, allowing us to investigate the atmospheric circulation associated with extreme events. To illustrate this we investigate the sea level pressure fields over the North Atlantic and Europe for four extreme rainfall months (Fig. 6). A one-to-one mapping between the monthly circulation patterns and rainfall does not exist and a similar circulation pattern could cause a range of rainfall, especially for a region as small as south east England. Nevertheless, the model patterns show circulation patterns that appear feasible in the real world and enable us to see a range of conditions that could lead to extreme rainfall over south east England.

The observed patterns from January 2014 and from January 1988, the two wettest Januarys in observations, are shown (Fig. 6a, b). The two are similar, showing patterns that would be expected to cause high rainfall as they have low pressure over the UK. Four examples of model Januarys with extreme rainfall are shown (Fig. 6c–f). Two of the extreme rainfall months show a similar pattern to observations, with low pressure centred near the UK (Fig. 6c, d), showing the model is able to simulate realistic flow patterns. This pattern represents storms repeatedly tracking across the North Atlantic and over south east England—as was observed in January 2014[1]. However, there is variation in the sea level pressure patterns leading to extreme monthly rainfall. For example, the circulation patterns in Fig. 6e, f project onto a negative North Atlantic Oscillation, not normally associated with wet and stormy conditions over the UK[26]. In these cases moist air from the subtropics moves northwards bringing rainfall to the UK. These examples serve to highlight that a variety of large-scale circulation patterns, some of them perhaps not yet realised, can drive UK regional extreme rainfall events. Future work could further investigate the dynamics of extreme and unprecedented rainfall and assess how the chance of regional rainfall extremes relates to the large-scale atmospheric circulation.

**Discussion**

We have used a large ensemble of climate simulations to assess the chance of unprecedented events in the current climate. The method could also be applied to other regions, timescales and climatic variables, e.g. hot summers. The model simulations also allow investigations into the causes and predictability of extreme events, but this is beyond the scope of this study. Future upgrades to even higher horizontal model resolution will allow better representation of orography and reduced model biases[19, 27, 28], allowing further application of this technique, which is likely to

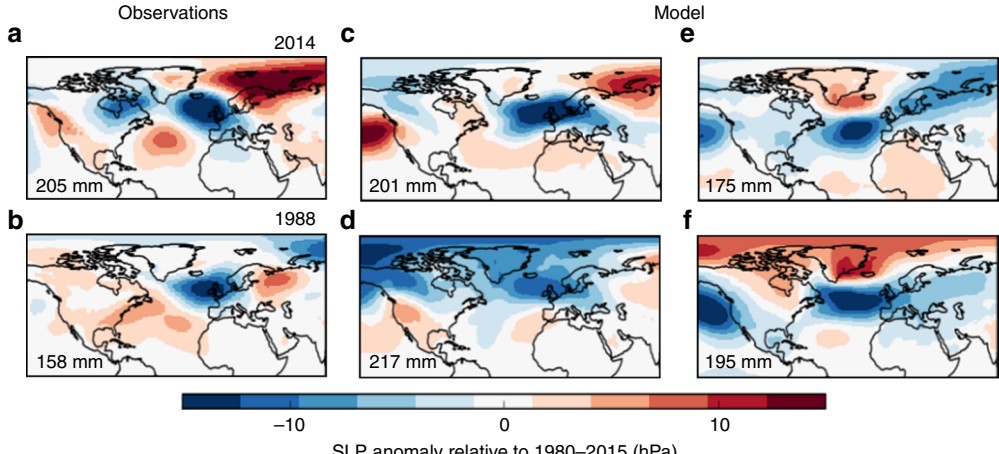

**Fig. 6** Observed and not yet realised climate states. **a, b** The sea level pressure anomaly fields (in hPa, relative to the January mean sea level pressure field) of the two observed Januarys with highest rainfall totals, 2014 and 1988, using the ERA data set[35]. **c–f** The sea level pressure anomalies of four extreme simulated Januarys, one of which **d** presents a potential new record rainfall scenario

become more important given the slow accrual of observations and the changing risks under climate change.

The large ensemble of model simulations, with two orders of magnitude more data than the observational record, allows us to better constrain the probability of extreme monthly rainfall. Furthermore, a model also allows the meteorological fields associated with the extreme events to be extracted. These could be used to diagnostically downscale to local rainfall using a fine scale regional model[29], and applied to hydrological models to evaluate the effects on river flow and potential impacts on society of such events.

With the current set of model simulations we cannot assess the risk of events greater than ~0.1% level using the ranking method as too few events fall into this category, leading to an overestimate of the probability at the rarest levels[30]. However, the model ensemble could potentially be increased to provide samples of rarer events if needed. As the number of ensemble members is increased the sampling uncertainty is reduced (compare *red* and *grey shading* in Fig. 5). This clearly shows the benefit of using a large ensemble of model simulations to constrain the risk of extreme rainfall events. However, our results using a single model are potentially over confident because they do not fully sample the structural uncertainties (Fig. 2). Further work repeating the analysis with simulations from other climate models is needed to understand the extent of this uncertainty.

So far we have considered only a single region finding the winter probability of unprecedented monthly rainfall in south east England is 7%, but the chance of unprecedented rainfall occurring in any region in England in a given year is higher. We extended the investigation to all regions of England where the model is consistent with the observations ('Methods'); the regions included are south east England, Midlands, East Anglia and north east England. There is a 34% probability of an unprecedented winter monthly rainfall total in at least one month in at least one region—it is therefore likely that we will see unprecedented winter rainfall within the UK in the next few years. These risk estimates are only valid in the current climate, future climate change is likely to alter the chances of extremes. This is a significant risk and could be used to inform decision makers on the likelihood and intensity of unprecedented rainfall events in the near future to protect the public, business and infrastructure from extreme rainfall and flooding.

## Methods

**Data**. The observational precipitation data used are from the Met Office NCIC station based records for the south east England[7]. Monthly data for 1980–2015

from the winter half of the year, October–March, is used. This provides 210 months of data.

Monthly model data are taken from 16 month long retrospective forecasts starting from November and 11 month long forecasts starting from May each year, 1980–2015. Forty ensemble members are available for each start time. The November start dates provide 2,800 simulations of the months between November and February (35 × 2 × 40: start dates × lead time × members). For October and March 1,400 months are available with only one lead time (35 × 1 × 40: start dates × lead time × members). For the May start dates 1,400 months are available for each calendar month from October to March. This provides 22,400 months in total (14,000 from the November start dates and 8,400 from the May start dates), equivalent to over 100 times the observed record.

**Model fidelity**. Fidelity tests are carried out to ensure that the model realistically represents the observed world. The mean, standard deviation, skewness and kurtosis are examined. To test the characteristics of the distribution proxy model time series the same length as the observational record are generated drawn from all years and ensemble members. We created 1,000 proxy time series for each of the 16 months from the model data, 16,000 in total. We calculated the mean, standard deviation, skewness and kurtosis of each proxy time series. The distribution of each of these measures is compared to the observed value. With only one realisation in the observed world there is a large degree of uncertainty in the observed distribution. For each measure the observational value must lie within 95% of the proxy model distribution for the modelled rainfall in the region to be deemed indistinguishable from the observations. The simulations from each start date (November and May) were also tested separately, and both passed the tests.

**Chance of monthly rainfall record**. The chance of unprecedented monthly rainfall in a given winter is found by calculating the percentage of model simulations in any calendar month exhibiting greater rainfall than the observed maximum. Comparing the modelled monthly rainfall totals with the observed record allows the rank risk curve to be plotted (Fig. 4, *red curve*). The 1% annual event probability is indicated with a horizontal line—the magnitude of such an event can be read off the curve. The uncertainty on the curve (Fig. 4, *red shading*) is estimated by creating subsamples of 14,000 model months 10,000 times, then calculating the 2.5–97.5% range of these subsamples. Our assessment of risks assumes that the climate is stationary over the period 1980–2015, and therefore may slightly underestimate risks associated with a changing climate[9]. However, splitting the period into two halves showed no significant difference in the risks.

A second method, using extreme value theory[22], is also used to calculate a risk curve (Fig. 4, *blue curve*). Block maxima of each model year are identified providing a distribution of over 2,000 values. A generalised extreme value distribution fitted to the data and used to calculate the return levels. Uncertainties (Fig. 4, *blue dashed curves*) are calculated by applying parametric bootstrapping 1,000 times and taking the 2.5–97.5% range[31].

In Fig. 5 extreme value theory is applied, using a generalised Pareto distribution fitted to observations[22]. Threshold fitting rather than block maxima due to the limited length of the record. The threshold of 118 mm (the 83.5th percentile) is used, leading to the inclusion 35 observed events. The same threshold is applied to the model data. Uncertainties are calculated by applying parametric bootstrapping 1,000 times, excluding fits with poor convergence, and then taking the 2.5–97.5% range. For the model the uncertainties are calculated once using the full data, and once sampling the distribution to the length of the observations.

To expand the study to cover all regions of England the model fidelity tests were carried out on six regions: south west England and south Wales, south east England, Midlands, East Anglia, north east England and north west England and north Wales. The model passes the fidelity tests in south east England, Midlands, East Anglia and north east England. The chance of an extreme in at least one month in at least one region is calculated by assessing every model simulation for the regions where the model passed the tests. The percentage of model simulations which include an unprecedented extreme in at least one month and region is calculated. We note that our study considers calendar months only and ignores any clustering of unprecedented events due to persistent circulation anomalies over a winter or over multi-annual to decadal timescales, e.g. caused by volcanic forcing[32], solar forcing[33] or ENSO[34].

**Code availability**. Due to intellectual property right restrictions, we cannot provide the source code or the documentation papers for HadGEM3-GC2. The Met Office Unified Model (MetUM) is available for use under licence. A number of research organisations and national meteorological services use the MetUM in collaboration with the Met Office to undertake basic atmospheric process research, produce forecasts, develop the MetUM code and build and evaluate Earth system models. For further information on how to apply for a licence, see http://www.metoffice.gov.uk/research/collaboration/um-partnership.

**Data availability**. The rainfall observations data used in this study is available from the National Climate Information Centre via https://www.metoffice.gov.uk/hadobs. The data used to produce the figures is available from the corresponding author for research use only.

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

## Acknowledgements

Development of the Met Office Hadley Centre's decadal climate predictions, the innovative scientific research that contributed to the NFRR, has been resourced through the MOHCCP, the NCIC, the Newton Fund, and SPECS. Development of the methodology was supported by the Newton Fund.

## Author contributions

D.M.S. and N.J.D. ran the ensemble hindcasts. A.A.S. and J.M.S. devised the original plan to use our ensemble hindcasts to examine unprecedented events. V.T. performed the analysis with input from N.J.D., A.A.S., D.M.S. and S.E.B.. S.B. advised on statistical methods. All the authors contributed to discussing the results and writing the paper.

## Additional information

**Competing interests:** The authors declare no competing financial interests.

