## [Peer Review file · Nature Communications]

Editorial Note: This manuscript has been previously reviewed at another journal that is not operating a transparent peer review scheme. This document only contains reviewer comments and rebuttal letters for versions considered at Nature Communications. Mentions of prior referee reports have been redacted.

Reviewers' comments:

Reviewer #1 (Remarks to the Author):

The paper presents a nice example of a use for a large ensemble of high resolution initialised climate simulations (retrospective forecasts) in simulating possible maximum rainfall amounts resulting from a known weather event. This shows that there is a high risk of unprecedented rainfall that has not already been seen in the current climate. The paper is a nice example of careful thinking and use of ensemble forecasts to tell us more about the current climate and potential events. However I have a few questions and a few revisions that I would like the authors to consider before I can recommend publication.

1. It is unclear in the current manuscript why the % chance of exceeding the current observed rainfall record is so much lower in southeast England compared to elsewhere (other regions of England and Wales). Perhaps the authors could discuss this more
2. Line 62: The simulations can only be used to calculate the risk of exceeding current monthly rainfall record totals if the climate is stationary (perhaps the authors could add something to this effect at this point)
3. Line 96: again expected exceedance assumes no climate change - make this clear
5. It would be useful to show the longer records for SE England on Figure 1 as well to see how the variability differs over the time period. Also add observed period used in Figure 1 to caption.

Reviewer #2 (Remarks to the Author):

Report for "High risk of unprecedented UK rainfall in the current climate"

Thompson et al. analyse the risk of a specific type of extreme precipitation over the UK, based on the ensemble seasonal prediction system of the UK MetOffice.

The overall findings of the authors are by-and-large sound and relevant. I am, however, wondering whether the study is innovative enough, and provides substantial enough results for Nature Communications.

My main concerns:

1. The authors claim that they develop "a novel use of large ensemble climate simulations to assess the risk of unprecedented events in the current climate". While I fully agree with the authors, that the approach is very useful to overcome the problem of short observational time series, I would like to point out that this approach has been applied already 11 years ago to assess the risk of extreme events in the Netherlands (van den Brink et al., 2005). This important paper has unfortunately not been cited by the authors.
2. I find the discussion of not-yet realised climate states honestly a bit shallow and not quite stringent (lines 109-126). The authors state that "although this particular pattern has not yet occurred in the real climate system, it may well do, and could drive even more extreme rainfall totals". This statement sounds rather naive to me, as it suggests that individual SLP patterns can directly be mapped onto precipitation patterns. It implies that since the rainfall simulated with this weather pattern is higher than the highest observed precipitation, this pattern cannot have been observed in reality. Have the authors checked this? I could well imagine that such an SLP pattern has been observed, even several times. But of course it could well be that each time this pattern

occurs, rainfall varies a bit - in the observations it might have been weaker by chance, in the model simulations higher values had occurred. Similarly, it could also be that the pattern causing the observed rainfall record could lead to even higher rainfall events. This discussion is very much linked to the question how strong regional rainfall is actually determined by the large-scale SLP patterns (this is essentially statistical downscaling), and of the sensitivity and specificity of the prediction (true/false positives, true/false negatives of record events).

Because this type of analysis has quite some potential, I am actually a bit puzzled why the authors do not focus more on this issue, and instead delete part of the validation (I would move much to the supplementary information).

3. I am a bit surprised that the authors assume a stationary climate in their assessment: a recent paper (Schaller et al., 2016, also cited by the authors) showed that anthropogenic global warming increased the risk of the 2013/14 events to occur in the UK. That is, it is likely that the risk in the beginning of the modelling period was lower than it is now. As a consequence, the assumption of stationarity may lead to an underestimation of the occurrence risk in present climate (the reported 8% within a year)! I would therefore suggest to carry out a non-stationary risk assessment (either by simply splitting the time interval into two halves, or by assuming a linearly increasing risk). Here the use of extreme value theory would help a lot to get robust estimates (see point 4).

4. The authors construct an antagonism between using long quasi-observational time series for assessing risk (their approach) and the use of extreme value theory to assess risk based on observational data. I agree that often the stationarity assumption of extreme value theory might not hold, and that the occurrence of very rare events might not easily be described by extrapolating from observed events. However, I find it rather odd that the authors do not combine the strength of their approach with extreme value theory: they take their ensemble simulations and purely empirically estimate risk (expressed in exceedance probabilities of a range of thresholds, Fig. 4). Applying extreme value theory to the ensemble situations could substantially reduce the associated uncertainties (and in this situation, no extrapolations would be required, the statistical theory would only be used to better constrain the rare "observations" in the tail). I would therefore urge the authors to recalculate Fig. 4 based on extreme value theory, including some uncertainty estimates (either simply by error propagation as carried out in the cited manuscript by Maraun et al., 2009; or even better by profile likelihood, Coles, 2001). In fact, to back up their skepticism of statistical extrapolation (as expressed mainly in lines 99-108) the authors should also compare their risk estimate with that based on observational data and extreme value theory.

Some minor issues:

line 57: "1050" shouldn't it be 2100 Januarys (as Jan is simulated twice in each forecast).

line 16/96: "therefore a new record is expected almost once per decade". No! Once a new record has occurred, the exceedance probability of course decreases (because the new threshold is higher).

References:

Coles, An introduction to statistical modeling of extreme values, Springer, 2001.

Maraun et al., The annual cycle of heavy precipitation across the United Kingdom: a model based on extreme value statistics, Int. J. Climatol. 29:1731-1744, 2009.

Schaller et al., Human influence on climate in the 2014 southern England winter floods and their impacts, *Nature Clim. Change* 6:627-634, 2016.

Van den Brink et al., Estimating return periods of extreme events from ECMWF seasonal forecast ensembles, *Int. J. Climatol.* 25:1345-1354, 2005.

Reply to Reviewers Comments on Manuscript NCOMMS-16-26743-T entitled "High risk of unprecedented rainfall in the current climate." By Thompson et al.

Reviewer #1:

The paper presents a nice example of a use for a large ensemble of high resolution initialised climate simulations (retrospective forecasts) in simulating possible maximum rainfall amounts resulting from a known weather event. This shows that there is a high risk of unprecedented rainfall that has not already been seen in the current climate. The paper is a nice example of careful thinking and use of ensemble forecasts to tell us more about the current climate and potential events. However I have a few questions and a few revisions that I would like the authors to consider before I can recommend publication.

We thank reviewer #1 for their positive comments and suggested revisions. We have used your comments to improve aspects of the manuscript.

1. It is unclear in the current manuscript why the % chance of exceeding the current observed rainfall record is so much lower in southeast England compared to elsewhere (other regions of England and Wales). Perhaps the authors could discuss this more

We give the chance of exceeding the current observed rainfall record in southeast England (7%) and the risk of a record in any of the regions of England (34%). The risk of a record in any region is higher as we have calculated the risk of a record in *any* region, not the risk in each region. Increasing the number of regions leads to the increased the risk. Emphasis has been added in the paragraph at lines 196-207 to clarify this.

2. Line 62: The simulations can only be used to calculate the risk of exceeding current monthly rainfall record totals if the climate is stationary (perhaps the authors could add something to this effect at this point)

We have altered the wording of lines 10, 43, and 68. We have tried to stress that we are assessing risk in the current climate only and assuming a stationary climate. We have also added a comment to line 204 stating the risk estimates are only valid in the current climate.

We have also tested the assumption of a stationary climate in the model by comparing the first and second half of the model data. We found the risk for both halves of the period are within the 95% uncertainty band of the full period (calculated from 16,000 subsamples of all years, as in Fig.4

of the manuscript). The two halves of the record are statistically indistinguishable from the full period; hence we do not include this figure in the manuscript though it is mentioned in the Methods (Line 245).

3. Line 96: again expected exceedance assumes no climate change - make this clear

We have added 'in the current climate' in line 104-105 to address this point.

4. It would be useful to show the longer records for SE England on Figure 1 as well to see how the variability differs over the time period. Also add observed period used in Figure 1 to caption.

The observational dataset that we are using, the Met Office NCIC record, is available from 1960 onwards. In our study we only use observational data from 1980 onwards to match the time period available from the model simulations. Above is the timeseries for all six months for the full period available – showing the record totals for each of the six months have occurred since 1980 (records are shown as circles). Including earlier observational data would not alter the risk calculations, so we prefer not to include the earlier observations in order to keep the figure as simple as possible, and directly comparable to the model simulations.

We do assess the risk of unprecedented rainfall using a longer timeseries, see lines 125-129. The longer timeseries covers 1880 to 2015, but includes fewer stations and covers a larger region. Despite the differences we find a similar risk - a 6% chance of at least one unprecedented monthly rainfall total in a given winter in south east England.

The caption of figure 1 has been updated to include the observed period.

Reviewer #2:

Report for "High risk of unprecedented UK rainfall in the current climate"

Thompson et al. analyse the risk of a specific type of extreme precipitation over the UK, based on the

ensemble seasonal prediction system of the UK Met Office.

The overall findings of the authors are by-and-large sound and relevant. I am, however, wondering whether the study is innovative enough, and provides substantial enough results for Nature Communications.

We thank you for the time you have put into reviewing our work and the insightful comments you have provided. Particular thanks for the suggestion of using extreme value theory, which we feel has significantly improved the paper.

My main concerns:

1. The authors claim that they develop "a novel use of large ensemble climate simulations to assess the risk of unprecedented events in the current climate". While I fully agree with the authors, that the approach is very useful to overcome the problem of short observational time series, I would like to point out that this approach has been applied already 11 years ago to assess the risk of extreme events in the Netherlands (van den Brink et al., 2005). This important paper has unfortunately not been cited by the authors.

Thank-you for pointing out this relevant and important reference, which we have now included at line 33. We have also included the earlier paper by the same authors, van den Brink et al., 2004. We have removed the word 'novel' from the abstract and elsewhere. However, due to model development and increases in supercomputer power over the last decade we are able to use a higher resolution model with a much larger set of model simulations. The model simulations we use cover the period 1981 to 2015, so will better cover the present day risk and may have a greater representation of the low frequency variability in the climate system than the 1987 to 2004 period available to van den Brink et al..

2. I find the discussion of not-yet realised climate states honestly a bit shallow and not quite stringent (lines 109-126). The authors state that "although this particular pattern has not yet occurred in the real climate system, it may well do, and could drive even more extreme rainfall totals". This statement sounds rather naive to me, as it suggests that individual SLP patterns can directly be mapped onto precipitation patterns. It implies that since the rainfall simulated with this weather pattern is higher than the highest observed precipitation, this pattern cannot have been observed in reality. Have the authors checked this? I could well imagine that such an SLP pattern has been observed, even several times. But of course it could well be that each time this pattern occurs, rainfall varies a bit – in the observations it might have been weaker by chance, in the model simulations higher values had occurred. Similarly, it could also be that the pattern causing the observed rainfall record could lead to even higher rainfall events. This discussion is very much linked to the question how strong regional rainfall is actually determined by the large-scale SLP patterns (this is essentially statistical downscaling), and of the sensitivity and specificity of the prediction (true/false positives, true/false negatives of record events). Because this type of analysis has quite some potential, I am actually a bit puzzled why the authors do not focus more on this issue, and instead delete part of the validation (I would move much to the supplementary information).

We agree that a 1 to 1 mapping of sea level pressure patterns to precipitation does not exist and have expanded the wording to make this clear (Lines 147-170). However, we do believe that the precise details of the circulation are very important in determining the total rainfall, and indeed

we cannot find observed patterns that exactly match those associated with extreme rainfall in the model.

We agree that analysing the driving patterns is potentially interesting, but a detailed understanding of how subtle differences affect the total rainfall is beyond the scope of our initial study. Hence, we have chosen to focus more on the validation of the method rather than analysis of the large scale dynamics at this point. Fig.6 is included primarily to show that the model rainfall can be driven by circulation patterns similar to those seen for the observed extremes, and the atmospheric conditions in the model appear physically plausible. We also use Fig 6 to make the important point that the large scale patterns, such as the North Atlantic Oscillation, cannot be assumed to relate directly to rainfall over a region as small as south east England.

3. I am a bit surprised that the authors assume a stationary climate in their assessment: a recent paper (Schaller et al., 2016, also cited by the authors) showed that anthropogenic global warming increased the risk of the 2013/14 events to occur in the UK. That is, it is likely that the risk in the beginning of the modelling period was lower than it is now. As a consequence, the assumption of stationarity may lead to an underestimation of the occurrence risk in present climate (the reported 8% within a year)! I would therefore suggest to carry out a non-stationary risk assessment (either by simply splitting the time interval into two halves, or by assuming a linearly increasing risk). Here the use of extreme value theory would help a lot to get robust estimates (see point 4).

Schaller et al. (2016) showed the risk has increased compared to a preindustrial base level – this change is included in both the observations and model simulation that we use. We do assume a stationary climate but only over the period from 1980 to 2015 – including climate change accrued prior to this period.

We have checked our assumption of stationarity by splitting the time period into two, as suggested. We found the risk for both halves of the period are within the 95% uncertainty band of the full period (calculated from 16,000 subsamples of all years, as in Fig.4 of the manuscript). The two halves of the record are statistically indistinguishable from the full period; which we now mention in the Methods (Line 245).

4. The authors construct an antagonism between using long quasi-observational time series for assessing risk (their approach) and the use of extreme value theory to assess risk based on observational data. I agree that often the stationarity assumption of extreme value theory might not hold, and that the occurrence of very rare events might not easily be described by extrapolating from observed events. However, I find it rather odd that the authors do not combine the strength of their approach with extreme value theory: they take their ensemble simulations and purely empirically estimate risk (expressed in exceedance probabilities of a range of thresholds, Fig. 4). Applying extreme value theory to the ensemble situations could substantially reduce the associated uncertainties (and in this situation, no extrapolations would be required, the statistical theory would only be used to better constrain the rare "observations" in the tail). I would therefore urge the authors to recalculate Fig. 4 based on extreme value theory, including some uncertainty estimates (either simply by error propagation as carried out in the cited manuscript by Maraun et al., 2009; or even better by profile likelihood, Coles, 2001). In fact, to back up their skepticism of statistical

extrapolation (as expressed mainly in lines 99-108) the authors should also compare their risk estimate with that based on observational data and extreme value theory.

Thank you for this useful suggestion. We have now applied extreme value theory to both the model and the observations. An extra co-author, Simon Brown, has been added to guide this additional analysis.

We have added a second curve to Fig.4, calculated using extreme value theory of the ensemble simulations. The two methods give highly consistent estimates of the risk of extremes.

We have also added a new plot (Fig.5). This shows the risk of absolute extreme rainfall totals, rather than as a proportion of the observed maximum. A plot of this form would be more useful for engineers and hydrologists. It also allows us to compare extreme value theory of the observations with the model simulations. When sampling the model simulations to 35 years (the length of the observations) the range of the uncertainties are similar to the observations. However, using the large ensemble of model simulations the uncertainties are much more tightly constrained compared to using observational data alone. This highlights the benefit of using the model simulations.

Thank-you for the suggestion of using extreme value theory, which we hope you will agree has improved the manuscript.

Some minor issues:

line 57: "1050" shouldn't it be 2100 Januarys (as Jan is simulated twice in each forecast).

Indeed – changed in text at line 62 (further updated to 4,200 Januarys due to the extra simulations now available). Thank-you for spotting that.

line 16/96: "therefore a new record is expected almost once per decade". No! Once a new record has occurred, the exceedance probability of course decreases (because the new threshold is higher).

Wording adjusted to 'a new record is likely to occur in the next decade' (line 105).

REVIEWERS' COMMENTS:

Reviewer #1 (Remarks to the Author):

The revised version is much improved. I am happy for it to be accepted after recourse to the following comments:

Line 14-15: I still don't think the abstract adequately captures why the probably goes up so much for all regions compared to the SE region I suggest change the text to "Expanding our analysis to some other regions of England and Wales the risk increases to a 34% chance of breaking a regional record somewhere each winter".

Line 137: I like the new use of extreme value analysis. Is it worth stating here though that the model actually provides a slightly lower chance of exceeding the record than the observations?

Line 306: typo – should be "from"

Reviewer #2 (Remarks to the Author):

The authors have done a great deal to improve the manuscript: they have placed it better in the context of existing literature, and they have complemented it by further analyses. I believe the results merit publication in Nature Communications.

There is just one remaining issue: the dichotomy between dynamical and statistical analysis is still too exaggerated. I agree with the authors that the assumptions underlying statistical extrapolation based on extreme value theory may not be fully valid for the climate system. But some statements are simply not true. E.g., one can also use extreme value theory with observations to gain insight into processes or precursors underlying the events, see, e.g., Toreti et al., 2010 for a composite based approach or Maraun et al., 2011 for a regression based approach. But of course the author's argument regarding the limited sample size in the observations is valid. The key advantage of extreme value theory in this context is that it provides a parametric description which reduces sampling uncertainties compared to non-parametric approaches. Therefore, using the dynamical and the statistical approach together (as done in the revision) is the method of choice. I simply ask to tune down the exaggerated dichotomy and to better work out the joint use of both approaches.

D. Maraun, T.J. Osborn and H.W. Rust: The influence of synoptic airflow on UK daily precipitation extremes. Part I: observed spatio-temporal relations, *Clim. Dynam.* 36(1-2), 261-275, 2011

A. Toreti, F.G. Kuglisch, E. Xoplaki, D. Maraun, H. Wanner and J. Luterbacher: Characterization of extreme winter precipitation in the Mediterranean and associated anomalous atmospheric patterns, *Nat. Hazards Earth Syst. Sci.* 10, 1037-1050, 2010

Reply to Reviewers Comments on Manuscript NCOMMS-16-26743-T entitled "High risk of unprecedented rainfall in the current climate." By Thompson et al.

Reviewer #1:

The revised version is much improved. I am happy for it to be accepted after recourse to the following comments:

Line 14-15: I still don't think the abstract adequately captures why the probably goes up so much for all regions compared to the SE region I suggest change the text to "Expanding our analysis to some other regions of England and Wales the risk increases to a 34% chance of breaking a regional record somewhere each winter".

We have added in 'somewhere', it helps clarify the point.

Line 137: I like the new use of extreme value analysis. Is it worth stating here though that the model actually provides a slightly lower chance of exceeding the record than the observations?

We have added a sentence commenting about the lower chance of exceedance shown by the model.

Line 306: typo – should be "from"

We have corrected this, thank-you for spotting.

Reviewer #2:

The authors have done a great deal to improve the manuscript: they have placed it better in the context of existing literature, and they have complemented it by further analyses. I believe the results merit publication in Nature Communications.

There is just one remaining issue: the dichotomy between dynamical and statistical analysis is still too exaggerated. I agree with the authors that the assumptions underlying statistical extrapolation based on extreme value theory may not be fully valid for the climate system. But some statements are simply not true. E.g., one can also use extreme value theory with observations to gain insight into processes or precursors underlying the events, see, e.g., Toreti et al., 2010 for a composite based approach or Maraun et al., 2011 for a regression based approach. But of course the author's argument regarding the limited sample size in the observations is valid. The key advantage of extreme value theory in this context is that it provides a parametric description which reduces sampling uncertainties compared to non-parametric approaches. Therefore, using the dynamical and the statistical approach together (as done in the revision) is the method of choice. I simply ask to tune down the exaggerated dichotomy and to better work out the joint use of both approaches.

We have changed line 152 to remove the incorrect comparison with statistical methods. We have also changed the wording around line 116 to include some advantages of a statistical method. We hope that this has tuned down the dichotomy between the two methods and instead highlights the advantages of the combined use.